# Patients with Methamphetamine Use Disorder Show Highly Utilized Proactive Inhibitory Control and Intact Reactive Inhibitory Control with Long-Term Abstinence

**DOI:** 10.3390/brainsci12080974

**Published:** 2022-07-24

**Authors:** Weine Dai, Hui Zhou, Arne Møller, Ping Wei, Kesong Hu, Kezhuang Feng, Jie Han, Qi Li, Xun Liu

**Affiliations:** 1CAS Key Laboratory of Behavioral Science, Institute of Psychology, Beijing 100101, China; daiweine15@mails.ucas.edu.cn (W.D.); zhouh@psych.ac.cn (H.Z.); 2Sino-Danish College, University of Chinese Academy of Sciences, Beijing 101408, China; 3Department of Nuclear Medicine and PET Center, Institute of Clinical Medicine, Aarhus University and University Hospital, 8200 Aarhus N, Denmark; arne@cfin.au.dk; 4Sino-Danish Center for Education and Research, Beijing 101408, China; 5Department of Psychology, University of Chinese Academy of Sciences, Beijing 100101, China; 6Beijing Key Laboratory of Learning and Cognition, School of Psychology, Capital Normal University, Beijing 100048, China; aweiping@gmail.com; 7Department of Psychology, Lake Superior State University, Sault St. Marie, MI 49783, USA; hkesong@gmail.com; 8Hebei Female Drug Rehabilitation Center, Shijiazhuang 050000, China; shengnvsuojiaoyuke@163.com (K.F.); hanjie121@126.com (J.H.)

**Keywords:** methamphetamine use disorder, proactive inhibitory control, reactive inhibitory control, sensation seeking

## Abstract

Methamphetamine use disorder (MUD) is a chronic brain disorder that involves frequent failures of inhibitory control and relapses into methamphetamine intake. However, it remains unclear whether the impairment of inhibitory control in MUD is proactive, reactive or both. To address this issue, the current study used the conditional stop-signal task to assess proactive and reactive inhibitory control in 35 MUD patients with long-term abstinence and 35 matched healthy controls. The results showed that MUD patients with long-term abstinence had greater preparation costs than healthy controls, but did not differ in performance, implying a less efficient utilization of proactive inhibitory control. In contrast, MUD patients exhibited intact reactive inhibitory control; reactive but not proactive inhibitory control was associated with high sensation seeking in MUD patients with long-term abstinence. These findings suggest that proactive and reactive inhibitory control may be two different important endophenotypes of addiction in MUD patients with long-term abstinence. The current study provides new insight into the uses of proactive and reactive inhibitory control to effectively evaluate and precisely treat MUD patients with long-term abstinence.

## 1. Introduction

Methamphetamine use disorder (MUD) is a chronic relapsing disorder according to the Diagnostic and Statistical Manual of Mental Disorders (5th edition, DSM-V), with frequent failures of self-control of methamphetamine intake despite methamphetamine abusers being well aware of the serious negative consequences. Theoretical models of addiction suggest that impaired inhibitory control is a key psychological mechanism underlying the development and maintenance of substance addiction [1,2]. Methamphetamine is one of the most commonly abused drugs worldwide [3], and the abuse of this drug has harmful effects on personal health and can even seriously endanger social security [4,5,6,7]. By the end of 2020, people who used methamphetamine accounted for the largest population of the more than 1.03 million synthetic drug users in China [8]. Therefore, investigating the inhibitory control of MUD patients is vital to understanding the mechanisms underlying addiction.

Inhibitory control can be categorized into two types: proactive and reactive. These types have different measurement methods and evaluation indexes, and their psychological and neural mechanisms are also different. The stop-signal task is one of the commonly used tasks for assessing inhibitory control that instructs participants to respond to go-signals (“Go” trials), but to withhold a response when a stop signal is presented (“Stop” trials). To study proactive inhibitory control, researchers have modified the classic stop-signal task by adding a condition in which participants do not need to prepare for inhibition. The index of proactive inhibitory control is preparation cost, i.e., the difference in response time between the prepared and unprepared conditions [9,10]. The index of reactive inhibitory control is the stop signal response time (SSRT), i.e., the time required for inhibition estimated based on the horse-race model [11,12]. According to the dual mechanisms of cognitive control [9,13], proactive inhibitory control is a goal-directed “early selection” process, triggered by predictive cues in the environment or internal signals, in which behavior is restrained in preparation for inhibition. In contrast, reactive inhibitory control is a stimulus-driven “late correction” process, triggered by external stop signals, in which behavioral responses are rapidly terminated.

Proactive inhibitory control may be an important protective factor that enables individuals with addiction to resist the temptation of addiction-related stimuli. For MUD patients, proactive inhibitory control is essential to their ability to refrain from using drugs when they experience unavoidable contact with methamphetamine-predictive cues or environments known to trigger use (i.e., preparing to reject an offer for drug use or to have a drug-free day), rather than relying only on reactive inhibitory control (i.e., inhibiting a response to seek out drug use) [14,15]. Therefore, the measurement of proactive inhibitory control may offer a more informative endophenotype for MUD [9]. Previous studies have found the presence of proactive inhibitory control in active alcohol-dependent individuals. A behavioral study using a modified stop-signal task provided supporting evidence, finding that as the probability of “stop” signals increased, active alcohol-dependent individuals utilized a higher level of proactive inhibitory control [15]. Using the classical stop-signal task to examine inhibitory control in active alcohol-dependent individuals and healthy controls (HCs), it was found that compared to HCs, active alcohol-dependent individuals also exhibited proactive inhibitory control but utilized proactive strategies less often, resulting in poorer inhibitory control performance (a lower “Go” accuracy) [16]. During abstinence, addicted individuals exhibited more adoption of proactive inhibitory control in response to addiction-related cues to engage in goal-directed behaviors. A study on cannabis addiction supported that both patients in abstinence treatment for cannabis use disorder and HCs utilized higher proactive inhibitory control to suppress responses to addiction-related stimuli compared to neutral stimuli [17]. This study also found that when craving was evoked in gamblers motivated to quit gambling, they utilized higher proactive inhibitory control in response to addiction-related cues and performed better on inhibitory control task than HCs and gamblers without evoked craving. However, research on whether MUD patients with long-term abstinence exhibit proactive inhibitory control to effectively accomplish goal-directed behaviors is lacking.

In MUD, previous studies of inhibitory control focused on reactive inhibitory control [18,19]. However, the results were inconsistent as to whether the reactive inhibitory control was impaired in MUD patients. Behavioral results in MUD patients with short-term abstinence (less than 15 days) found that their SSRT was longer than that of HCs, which represents a worse reactive inhibitory control [20,21]. A behavioral study of MUD patients with long-term abstinence (more than 15 days) found intact reactive inhibitory control of MUD [22]. Previous studies of other stimulant addictions found that reactive inhibitory control might progressively recover with prolonged abstinence [23,24]. Therefore, we speculated that this inconsistency in results may be due to variance in the duration of abstinence.

In addition, one of the key factors of impaired inhibitory control in MUD patients is excessive sensation seeking. Previous studies found that inhibition was associated with sensation seeking at the self-report and cognitive/behavioral levels and was often implicated in externalizing behaviors, such as drug addiction [25,26,27]. However, addiction research that explores the relationships among the different subdimensions of inhibitory control, proactive inhibitory control, reactive inhibitory control and sensation seeking is lacking.

The goals of the current study were to investigate the function of reactive and proactive inhibitory control in MUD patients with long-term abstinence as well as the relationship between sensation seeking (a personality trait that renders individuals susceptible to MUD) and inhibitory control. The conditional stop-signal task was applied to measure both reactive and proactive inhibitory control, and the sensation-seeking personality trait was measured using the Sensation Seeking Scale. We hypothesized that similar to abstinence in people with cannabis use disorder or pathological gambling problem, MUD patients with long-term abstinence would utilize more proactive inhibitory control to accomplish goal-directed behaviors than HCs. Based on previous literature indicating that reactivity inhibitory control could be recovered with long-term abstinence [23,24], we hypothesized that MUD patients with long-term abstinence would show intact reactive inhibitory control. Furthermore, we predicted that the higher the sensation seeking of MUD patients with long-term abstinence was, the poorer their inhibitory control would be.

## 2. Materials and Methods

### 2.1. Participants

All participants were examined by the physicians and psychologists of rehabilitation centers using standardized scales and interview techniques. The inclusion criteria for all participants were as follows: age 18 or older; no history of or current neurological disorders or severe mental disorders; no chronic somatic diseases or brain injury; and no substance use disorders (e.g., heroin, cannabis, cocaine, alcohol or nicotine) or behaviors (e.g., gambling or internet) according to the DSM-V.

Thirty-five MUD patients according to the DSM-V diagnostic criteria from two compulsory addiction rehabilitation centers in Hebei Province, China, were included in this study. All the MUD patients were primarily monosubstance-dependent users and met the severe criteria. These MUD patients had abstained from methamphetamine intake for an average of 264.89 (SD = 167.35) days and had previously taken an average dose of 265.57 (SD = 428.87) grams. In abstinence treatment, drug rehabilitation centers help MUD patients suppress/stay away from addiction-related stimuli by conducting expert lectures, popularizing science and speaking on behalf of successful drug rehabilitation, which benefits MUD patients’ awareness of the dangers of drugs and their initiative to stay away from drugs. And each subject’s lifetime methamphetamine dosage was estimated by recalling the sum of the daily dosage multiplied by the number of days used in each month from the first to the most recent methamphetamine use.

Thirty-five HCs from nearby communities were recruited through referrals from center staff, word-of-mouth, and print advertisements. They were matched for age, gender and education level with the MUD patients. The HCs were carefully assessed for a history of substance use and were excluded from the study if they met any criteria for substance use disorders.

All participants received a participation incentive of 45 Chinese yuan (CNY) and were transferred to MA patients’ savings cards and to HCs’ Alipay accounts. The study was approved by the Institutional Review Board of the Institute of Psychology, Chinese Academy of Sciences, and conducted in accordance with the Declaration of Helsinki. Demographic characteristics of the participants are presented in Table 1.

### 2.2. Measurements

#### 2.2.1. Questionnaires

Depression. The Chinese version of the Beck Depression Inventory—version 2 (BDI-II) [28] was used to measure the severity of depression. This scale is one of the most widely used self-rating scales for depressive symptoms and contains 21 items; the total score is the sum of all items.

Trait anxiety. The Chinese version of the State-Trait Anxiety Inventory—T (STAI-T) [29] was used to measure trait anxiety. The STAI consists of two subscales: the State Anxiety Inventory and the Trait Anxiety Inventory. In this study, only the Trait Anxiety Inventory, which is widely used in clinical research to evaluate the relatively stable emotional states of various individuals or groups, was used to measure the trait anxiety of the participants. This inventory contains 20 items, and the total score is the sum of all items.

Sensation seeking. The Chinese version of the Sensation Seeking Scale [30] was used to measure sensation seeking. This scale includes four subscales with 10 items each: Thrill and Adventure Seeking measures the desire to participate in activities that produce intense feelings and experiences; Experiment Seeking measures the seeking of sensations and experiences through the mind and the senses, as in music, art, travel, and social nonconformity and unconventionality; Disinhibition of Desire measures the desire for social and sexual disinhibition, as expressed by seeking variety in social drinking, partying, and sexual partners; and Boredom Susceptibility measures intolerance for repetitive experience or predictable and unexciting people. Summing all 40 items yields an overall sensation-seeking score.

All of the questionnaires were presented in and data were collected by the Python-based software PsychoPy2 (Open Science Tools Ltd., Nottingham, UK).

#### 2.2.2. Task

Inhibitory control was assessed by the conditional stop-signal task (C-SST), a paradigm modified from tasks applied in Chikazoe et al. [31] and Zheng et al. [32] and followed by a consensus guide to the stop-signal task [33]. The paradigm presentation and data requisition were conducted with the Python-based software PsychoPy2 on a computer. Prior to the formal task, each participant performed a pretest task including 50 trials to determine their baseline reaction time (RT) for the formal task. In each trial, a fixation was presented for 500 ms, followed by an 800-ms presentation of a context cue (blue circle) which indicated that the “Stop” signal would not appear. Then, a white arrow, which served as both the “Go” stimuli and target, was presented for 1200 ms. Participants were required to determine the direction the arrow pointed (up or down) and press the corresponding key (“F” or “J” on the keyboard) with their left or right index finger as quickly as possible. The assignment of response mappings to arrow directions was counterbalanced across participants. The baseline RT of participants was set as the 90th percentile (in ascending order) of the participant’s own RT distribution.

The formal task included two certain blocks and two uncertain blocks (Figure 1). Each block included 60 trials, and the order of blocks was counterbalanced across participants. In the certain blocks, the task was the same as that in the pretest task on Certain-Go trials, but the arrow randomly changed from white to red (i.e., the “Stop” signal) on 40% of trials (Ignore-Go trials). The context cues in the certain blocks were blue circles, which indicated that participants should ignore the “Stop” signal and respond to the direction of the arrow. After each trial, feedback was presented for 800 ms, informing participants that their responses were “correct” (pressed the correct key), “wrong” (pressed the wrong key), or “slow” (pressed the right key but RT > baseline RT). In the uncertain blocks, participants still performed the same button-press task, except that the arrow randomly changed from white to red (i.e., the “Stop” signal) on 40% of trials (“Stop” trials). The context cues in the uncertain blocks were yellow circles, which indicated that participants should inhibit their ongoing response when they saw the “Stop” signal. The feedback on “Stop” trials was “correct” (no key press), or “wrong” (key press, regardless of whether it was correct). The time delay between the “Go” stimuli and the “Stop” signal, defined as the stop-signal delay (SSD), was initially 300 ms and varied among trials. In certain blocks, the SSD was randomly selected in the range of 50–500 ms. In uncertain blocks, this was done through a tracking procedure that converged on the critical SSD, where participants successfully inhibited responses on 50% of the “Stop” trials. After a successful “Stop” trial, the next “Stop” trial was made more difficult by adding 50 ms to the SSD; if inhibition failed, then the next “Stop” trial was made less difficult by subtracting 50 ms from the SSD. The range of the SSD in uncertain blocks was 50–500 ms. To prevent strategic slowing, participants received performance feedback at the end of each block. If accuracy was less than 85%, the feedback was “Your response accuracy is relatively low; please increase your accuracy!”; if the mean RT was slower than the baseline RT, the feedback was “Your time to response is relatively long; please speed up!”; otherwise, the feedback was “Your performance is excellent; please keep it in that way.” Before the formal test started, each participant underwent 10 training trials to become acquainted with the tasks in each type of block.

#### 2.2.3. Data Analysis

Data were analyzed using Statistical Package for the Social Sciences 26.0 statistical software (SPSS, Inc., Chicago, IL, USA) and MATLAB R2017a (The MathWorks, Inc., Natick, MA, USA). For the demographic and questionnaire data, independent sample t-tests were used to compare group differences (HC group vs. MUD group).

For the behavioral data on the C-SST, each index of cognitive ability was calculated. Preparation cost, the index of proactive inhibitory control, was estimated by subtracting the mean Certain-Go RT from the mean Uncertain-Go RT for each participant. This difference reflected the delay in response when individuals were required to anticipate and prepare for inhibiting an ongoing response [31]. The critical SSD was determined from the tracking procedure as the mean of the peak and valley values of SSDs in “Stop” trials. The critical SSD represented the time delay corresponding to a 50% success rate in inhibiting a response in the uncertain blocks [34]. The SSRT, the index of reactive inhibitory control, was calculated based on the horse-race model and estimated by subtracting the critical SSD from the correct mean Uncertain-Go RT for each participant [35,36]. The post-signal slowing (PSS) effect, the index of performance monitoring, was estimated by subtracting the mean Go RT following an Uncertain-Go trial from the mean Go RT following a “Stop” trial for each participant [37].

The group differences in baseline RT, preparation cost, critical SSD, SSRT and PSS were analyzed with independent t tests. Two 2 (group: MUD vs. HC) × 4 (context: Certain-Go vs. Ignore-Go vs. Uncertain-Go vs. Stop) mixed-design analyses of variance (ANOVAs) were used to assess RT and accuracy. The Greenhouse–Geisser epsilon correction was used in case the assumption of sphericity was violated, and the Bonferroni procedure was applied for posthoc comparisons. Two non-parametric two-way ANOVAs (Scheirer–Ray–Hare tests) were used to investigate the effects of group and gender on preparation cost and SSRT by R. Pearson correlation analyses were conducted for all participants, HCs and MUD patients to investigate the relationship between proactive and reactive inhibitory control in the dual mechanisms of cognitive control as well as the relationship between inhibitory control and sensation seeking. The significance threshold was a *p*-value of 0.05 after correction by the false discovery rate (FDR).

## 3. Results

### 3.1. Questionnaire Results

The results of the questionnaires are presented in Table 2. There were no differences in depression and anxiety between MUD and HC participants (*ps* > 0.05). As expected, the total scores on the personality traits of sensation seeking showed that patients in the MUD group scored higher than the HC group (*ps* < 0.05). Specifically, MUD patients showed stronger sensation seeking in the Disinhibition of Desire (*p* < 0.05) and Thrill and Adventure Seeking subscales (*p* < 0.01).

### 3.2. C-SST Results

The behavioral measures are summarized in Table 3 and Figure 2. The MUD group had a significantly longer preparation cost than the HC group (*p* < 0.01), but there were no group differences in the baseline RT, critical SSD, SSRT or PSS (*ps* > 0.05) (Figure 2A).

The mixed-design ANOVA on RT found a significant main effect of context, *F* (3, 204) = 213.11, *p* < 0.001, $\eta_{p}^{2}$ = 0.76 (Figure 2B). Posthoc comparisons revealed that RT on trials increased in the following order: Certain-Go, Ignore-Go and Fail-to-Stop (*ps* < 0.001). It suggested that the slowing in SSRT indeed reflected inhibitory control rather the differences in the properties of the “Stop” signal itself. The interaction between group and context was significant, *F* (3, 204) = 6.20, *p* < 0.01, $\eta_{p}^{2}$ = 0.08. Posthoc comparisons revealed no significant group differences due to different contexts (*ps* > 0.05). In the HC group, Certain-Go RT and Ignore-Go RT < Fail-to-Stop RT < Uncertain-Go RT (*ps* < 0.001); there was no significant difference between Certain-Go RT and Ignore-Go RT (*p* > 0.05). In the MUD group, the RT on the trials increased in the following order: Certain-Go, Ignore-Go and Fail-to-Stop (*ps* < 0.001).

The mixed-design ANOVA on accuracy found a significant main effect of context, *F* (3, 204) = 2626.24, *p* < 0.001, $\eta_{p}^{2}$ = 0.98 (Figure 2C). Posthoc comparisons revealed that, the accuracies on the Certain-Go and Uncertain-Go trials > the accuracy on the Ignore-Go trials > the accuracy on the Stop trials (*ps* < 0.01); there was no significant difference in accuracy on the Certain-Go and Uncertain-Go trials (*p* > 0.05). The accuracy on the Stop trials was approximately 50% and significantly different from the other three conditions (*p* < 0.001), indicating that the SSD was manipulated effectively. No significant main effect of group or interaction of group and context was found.

To investigate gender differences in the inhibitory control of MUD, two non-parametric two-way ANOVAs of preparation cost and SSRT were applied in current study. For preparation cost, we found a significant main effect of group, *H* (1) = 7.25, *p* = 0.007; however, the main effect of gender, *H* (1) = 0.04, *p* = 0.838 and the interaction between group and gender were not significant, *H* (1) = 1.98, *p* = 0.159. For SSRT, we found a significant main effect of gender, *H* (1) = 3.87, *p* = 0.049, wherein male was worse than female; however, the main effect of group, *H* (1) = 0.07, *p* = 0.796, and the interaction between group and gender were not significant, *H* (1) = 0.51, *p* = 0.473.

### 3.3. Correlation Analysis Results

Regarding the relationship between proactive and reactive inhibitory control in the dual mechanisms of cognitive control, Pearson correlation analysis indicated that preparation cost was positively correlated with SSRT in all participants (*r* = 0.38, *p* = 0.005; FDR-corrected) (Figure 3A) and MUD (*r* = 0.63, *p* < 0.001, FDR-corrected) (Figure 3C). There was no significant correlation between preparation cost and the SSRT in the HC group (*r* = 0.07, *p* = 1.000; FDR-corrected) (Figure 3B).

Regarding the relationship between inhibitory control and sensation seeking, Pearson correlation analysis showed that the SSD was negatively correlated with total Sensation Seeking Scale scores in the MUD group (*r* = −0.44, *p* = 0.038; FDR-corrected) (Figure 4C), indicating that stronger sensation seeking was accompanied by poorer reactive inhibitory control. Pearson correlation analysis indicated that inhibitory control was not correlated with sensation seeking in all participants or in the HC group (*ps* > 0.05, FDR-corrected) (Figure 4A,B).

Additionally, a relationship between sensation seeking and the severity of addiction was found in the MUD group. Pearson correlation analysis indicated that the Disinhibition of Desire subscale of the Sensation Seeking Scale was negatively correlated with severity measured by DSM-V in the MUD group (*r* = 0.42, *p* = 0.045; FDR-corrected) (Figure 5), indicating that stronger sensation seeking was accompanied by poorer reactive inhibitory control.

## 4. Discussion

The present study used the conditional stop-signal task to investigate changes in inhibitory control. To our knowledge, this is the first behavioral study to simultaneously compare proactive and reactive inhibitory control in MUD patients with long-term abstinence and to explore the relationship between the two types of inhibitory control and the personality trait of sensation seeking. The results suggest that MUD patients with long-term abstinence utilized more proactive inhibitory control than HCs and that their reactive inhibitory control was intact. No significant differences were found in “Go” processing and PSS effects between MUD patients with long-term abstinence and HCs, ruling out the possibility that the inefficient proactive inhibitory control in MUD patients may emerge due to executive functioning and performance monitoring. Additionally, we found that inhibitory control in MUD patients with long-term abstinence was associated with sensation seeking.

### 4.1. MUD Patients with Long-Term Abstinence Utilized More Proactive Inhibitory Control than HCs

The current study found that MUD patients with long-term abstinence had a larger preparation cost than HCs, while their performance (accuracy of response to the “Go” stimulus) was the same. Proactive inhibitory control is a goal-driven “early selection” process, preparing for the possibility of behavioral suppression [9,13]. MUD patients with long-term abstinence need to utilize goal-driven proactive inhibitory control to resist the temptation and craving induced by unavoidable exposure to methamphetamine-related cues or environments [17]. Thus, in the C-SST, MUD patients with long-term abstinence prioritized accuracy over speed by delaying their response until additional contextual information was gathered.

The group difference in proactive inhibitory control between MUD patients with long-term abstinence and HCs may be due to the following reasons. First, the difference in abstinence motivation may have led to the group difference in proactive inhibitory control. Previous research found that active alcohol-dependent patients exhibited less utilization of proactive inhibitory control, however, resulting in poorer performance [16]. This may reflect a tendency to be willing to enjoy the experience of addiction-related stimulus and suggests that proactive inhibitory control may be an important indicator of and a susceptibility factor for the development and maintenance of addiction. However, patients during abstinence are highly motivated to utilize proactive inhibitory control to suppress the temptation of addiction-related stimuli and thus control addiction-related behaviors [17]. Patients with cannabis use disorder during abstinence utilize higher proactive inhibitory control to inhibit the processing of cannabis-related cues; when the craving was evoked, gamblers who were motivated to quit utilized higher proactive inhibitory control to suppress their craving and attempt to control gambling behaviors [17]. Similarly, in the present study that used the C-SST, MUD patients with long-term abstinence utilized more proactive inhibitory control to prepare for “Stop” signals, ensuring that they were able to inhibit their action when “Stop” stimulus was presented. Moreover, we also found that MUD patients with long-term abstinence traded off speed (preparation cost) for maintaining the same level of performance (the accuracy) as HCs, indicating that the processing of proactive inhibitory control was less efficient and may be related to aberrant brain function caused by long-term methamphetamine use. Previous studies of inhibitory control have found that right inferior frontal gyrus (rIFG), anterior cingulate gyrus and prefrontal cortex activation were decreased in MUD patients who had been abstinent for more than 3 weeks during the Stroop task [18,38]. Moreover, the attenuated rIFG activation during proactive inhibitory control found in cocaine use disorders may imply that the blunted inhibitory ability of CUD participants stems from deficits in initial attention and detection of potential stopping targets [39]. However, there is a lack of brain imaging studies of proactive inhibitory control of MUD, as more fMRI studies on this issue are needed in the future.

Second, abstinence may increase the salience of inhibition anticipation in patients with addiction, reflecting a stronger allocation of top-down attention control [40]. Patients with addiction may exercise proactive inhibitory control more strongly, increasing their use of more inhibitory, restrictive, and/or avoidant behavioral coping strategies [41,42]. In contrast to HCs, MUD patients with long-term abstinence have undergone treatments for abstinence that emphasize inhibition or avoidance of addiction-related stimuli. By reinforcing individual proactivity, inhibitory control could be enhanced in patients with addiction. MUD patients with long-term abstinence in the present study may have consciously utilized proactive inhibitory control to accomplish the behavior; alternatively, their framing context of inhibitory control may have been more salient than that of HCs, with stronger top-down attentional control over inhibition-related stimuli, utilizing more preparation to confront “Stop” signals.

### 4.2. MUD Patients with Long-Term Abstinence Showed Intact Reactive Inhibition Control

In the current study, not only the SSRT but also the performance (accuracy) of MUD patients with long-term abstinence did not differ from that of HCs. The core features that distinguish reactive inhibitory control from proactive inhibitory control are a stimulus-driven “late correction” process and a rapid cancelation of behavioral responses upon the sudden appearance of a “Stop” signal [9,13]. Worse reactive inhibitory control may contribute to impulsive behaviors that reflect an inability to immediately and rapidly terminate addiction-related behavioral responses to resist unexpected temptations or overcome impromptu cravings [21,43,44,45,46].

As expected, no group differences were found in reactive inhibitory control. A possible explanation may be that the prolonged abstinence facilitated the functional recovery of reactive inhibitory control. Previous behavioral results in MUD patients have found that their reactive inhibitory control was worse with shorter periods of abstinence [20,21], but intact with longer periods of abstinence [22]. This inconsistency suggests that the reactive inhibitory control of MUD patients may gradually recover with prolonged abstinence. Another possible explanation for the intact reactive inhibitory control of MUD patients with long-term abstinence may be related to detoxification. Many studies have found that acute intoxication [47,48] and chronic intoxication [49,50] from drug abuse can impair reactive inhibitory control. Abstinence diminishes the effect of intoxication, thus restoring reactive inhibitory control [23,24].

### 4.3. Inhibitory Control of MUD Patients with Long-Term Abstinence Was Associated with Sensation Seeking

As hypothesized, investigations into the personality traits related to addiction revealed that MUD patients with long-term abstinence exhibited higher levels of sensation seeking than HCs. Moreover, in MUD patients, sensation seeking was associated with reactive inhibitory control, but not proactive inhibitory control. Sensation seeking is a personality trait as well as a potential endophenotype for various addictive behaviors [51,52] and correlates with addiction severity [53], which is consistent with the current results. Previous studies have suggested that individuals with addiction exhibit a higher level of sensation seeking than HCs and experience a greater desire to seek euphoria through stimulant consumption [54,55]. According to addiction theory, excessive craving for addiction-related stimuli leads to reduced inhibitory control in MUD patients, and this impairment of reactive inhibitory control generalizes to general stimuli as the severity increases [56]. Evidence from empirical studies suggests that individuals with addiction have higher levels of sensation seeking and worse reactive inhibitory control [55,57]. In the current study, in MUD patients with long-term abstinence, the personality trait of sensation seeking scores was negatively associated with the SSD, suggesting that reactive inhibitory control needs to be accomplished under easier conditions for MUD patients. This may indicate that reactive inhibitory control mediates the relationship between sensation seeking and addictive behaviors. However, proactive inhibitory control strategies place more emphasis on an individual’s top-down, goal-oriented ability to mobilize cognitive resources to inhibit current behaviors as well as an individual’s proactivity in the current state, resembling self-control, which indirectly influences addictive behaviors [58]. Thus, no correlation was found between sensation seeking and proactive inhibitory control.

Furthermore, the current study indicated that the processing of proactive and reactive inhibitory control was independent and complementary for MUD patients with long-term abstinence. On the one hand, the independence of the two types of inhibitory control was reflected in the following aspects. First, while there was a group difference in proactive inhibitory control, reactive inhibitory control did not show a group difference, suggesting that abnormalities in the two types of inhibitory control can occur independently. Second, no significant correlation was found between proactive and reactive inhibitory control in the HCs, suggesting that the two types of inhibitory control may be implemented independently in this population. Third, in MUD patients with long-term abstinence, a correlation was found between only reactive inhibitory control and sensation seeking, whereas no such correlation was found for proactive inhibitory control, further indicating the independence of the two types of inhibitory control. Fourth, previous neuroimaging studies found that proactive inhibitory control specifically activates the bilateral superior parietal lobule, reflecting the top-down influence over motor control [40,59]. While reactive inhibitory control is uniquely associated with the right dorsolateral prefrontal cortex, ventrolateral prefrontal cortex and anterior supplementary motor area, reflecting stimulus-driven attention and reprogramming from action initiation to cancellation [40,60]. On the other hand, proactive and reactive inhibitory control are also complementary. In current study, for MUD patients with long-term abstinence, those with worse reactive inhibitory control were more likely to utilize proactive inhibitory control, which may be a compensatory inhibitory processing mechanism. In addition, consistent with the current study, previous studies found that proactive and reactive inhibitory control share neural circuits involved in blocking and braking motor output, including the inferior frontal gyrus, supplementary motor area, subthalamic nucleus, and striatum [9,40]. The present study provides empirical support for the theory of dual mechanisms of cognitive control [9,13] from the perspective of MUD patients with long-term abstinence.

### 4.4. Limitations and Prospect

One limitation of this study is the lack of a specific division to distinguish short-term and long-term abstinence in addiction population. Previous studies found that proactive and reactive inhibitory control in patients with addiction was altered depending on the duration of abstinence. However, due to the lack of active MUD patients, it is difficult to investigate the dynamics of or distinction between proactive and reactive inhibitory control among addiction patients with active use, short-term abstinence, and long-term abstinence that are addicted to the same substance. Thus, the characteristics of proactive and reactive inhibitory control in MUD patients warrant investigation with a longitudinal study. A second limitation is that the research paradigm lacks addiction-related cues. As addiction cues provide a specific context for addicted patients, an examination of their inhibitory control to addiction-related stimuli is needed. Therefore, addiction-related cues should be added to the paradigm in future studies to examine differences in proactive and reactive inhibitory control over addiction-related and non-addiction-related cues in MUD patients. Third, the current study did not find gender differences in inhibitory control of MUD. Although gender differences are observed in drug response, addictive behavior and mental health of MA patients [61,62,63], no gender differences in proactive inhibitory control has been investigated in previous studies of MUD. In the current study, we did not find gender differences in inhibitory control of MUD, neither in proactive nor in reactive inhibitory control. This is consistent with previous studies of substance addiction using SST, where behavioral outcomes did not find significant interactions of group and gender [22,64,65]. However, the sample size of the current study is relatively small, so more replicated studies as well as studies with expanded samples are needed in the future to further examine the reliability of this conclusion.

The difficult recovery of proactive inhibitory control may be an important reason for the high relapse rate of addiction. Previous studies have found that reactive inhibitory control could be recovered with long-term abstinence therapy [22,23,24], but the relapse rate after abstinence remains high [66,67]. The current study found no difference in reactive inhibitory control between MA with long-term abstinence and HCs, but proactive inhibitory control remained a group difference. It suggests that long-term abstinence treatment may restore reactive inhibitory control, but the improvement in proactive inhibitory control is still insufficient, which may be an important factor for the high relapse rate. These results suggest that the future treatment of addiction should be further developed to treat proactive inhibitory control. Through training proactive inhibition, it could be made become the habitual response of MA patients in order to reduce the relapse rate.

## 5. Conclusions

The present study provides evidence of the behavioral characteristics of proactive and reactive inhibitory control in MUD patients with long-term abstinence, suggesting that the effectiveness of proactive inhibitory control in MUD patients with long-term abstinence was low, but that their reactive inhibitory control remained intact, possibly related to recovery after long-term abstinence. In addition, a correlation between sensation seeking and inhibitory control was found in MUD patients with long-term abstinence. These results may provide evidence that proactive and reactive inhibitory control are critical endophenotypes and viable targets in abstinent patients with addiction and that high sensation seeking is a risk factor for addiction, but these conclusions are preliminary. Additionally, recovery from MUD may also require interventions beyond abstinence, such as cognitive-behavioral therapy, motivational enhancement therapy and contingency management [68,69], depending on the patient’s behavioral characteristics and personality traits.

## Figures and Tables

**Figure 1 brainsci-12-00974-f001:**
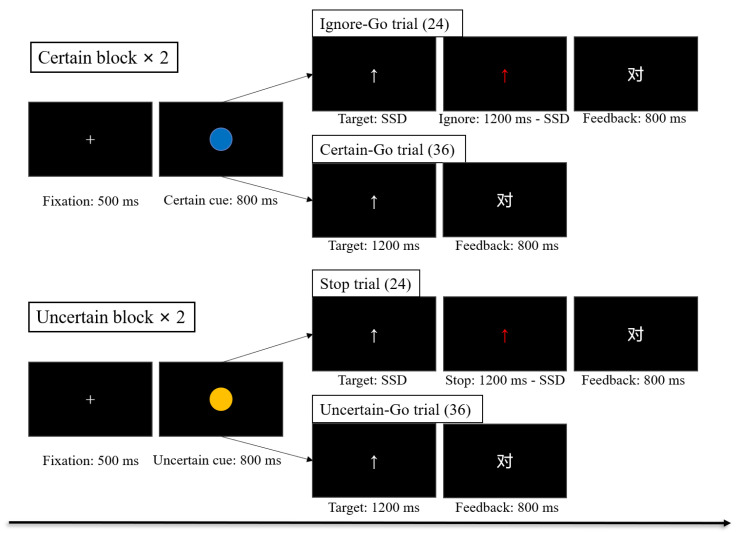
Illustration of the C-SST design. Both certain and uncertain blocks contained red arrows as the “Stop” signal in 40% of the trials. Note: SSD = the stop-signal delay. The feedback screen shows an example of a Chinese character that means “correct”.

**Figure 2 brainsci-12-00974-f002:**
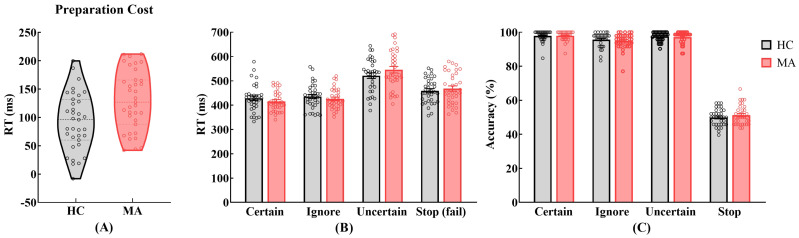
Behavioral results on the C-SST. The black border, gray-filled violin plot and bars represent the HC results, and the red border, pink-filled violin plot and bars represent the MUD results. The circles within each violin plot and bar represent data for individuals datapoints. (**A**) Violin plots reveal that the preparation cost in the MUD group was significantly slower than that in the HC group (*p* < 0.01). The upper, middle, and lower dashed lines within each violin plot represent the upper, middle, and lower quartiles, respectively. (**B**) The grouped bar chart presents the RTs under the four contexts (Certain-Go, Ignore-Go, Uncertain-Go and Fail-to-Stop) for the HC and MUD groups. (**C**) The grouped bar chart represents the accuracy in the four contexts (Certain, Ignore, Uncertain and Stop) for the HC and MUD groups.

**Figure 3 brainsci-12-00974-f003:**
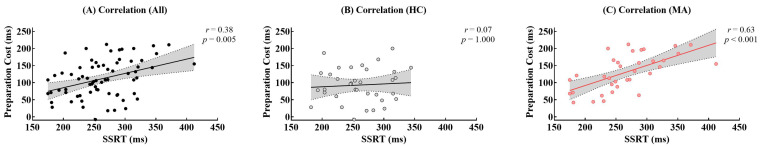
Scatter plots of preparation cost and SSRT with regression lines and 95% CIs in all participants (**A**), and the HC (**B**) and the MUD groups (**C**). Note: CI, confidence interval; *r*, Pearson correlation coefficient; *p* values corrected by FDR.

**Figure 4 brainsci-12-00974-f004:**
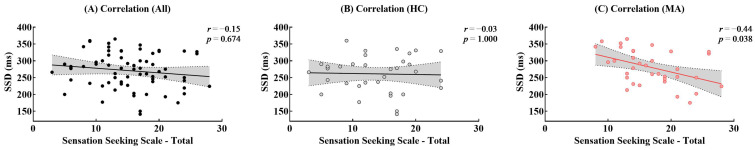
Scatter plots of SSD and total Sensation Seeking Scale scores with regression lines and 95% CIs in all participants (**A**) and the HC (**B**) and MUD groups (**C**). Note: *r*, Pearson correlation coefficient; *p*-values corrected by FDR.

**Figure 5 brainsci-12-00974-f005:**
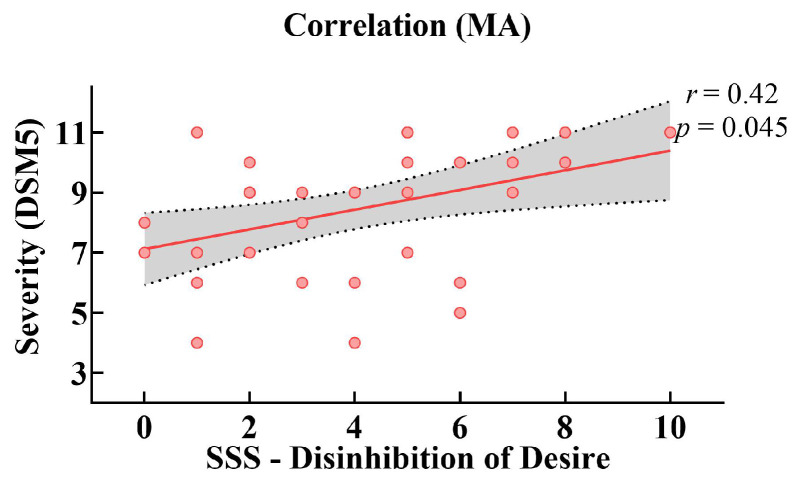
Scatter plot of the Disinhibition of Desire subscales of Sensation Seeking Scale and MUD severity with regression lines and 95% CIs in the MUD group. The severity of MUD was measured by DSM-V. Note: *r*, Pearson correlation coefficient; *p* values corrected by FDR.

**Table 1 brainsci-12-00974-t001:** Demographic characteristics of HCs and MUD patients.

	HC (N = 35, Female = 17)		MUD (N = 35, Female = 19)		*t* Value	*p* Value
	Mean	SD	Mean	SD		
Age (years)	29.43	6.66	27.34	6.95	*t* (68) = 1.28	0.204
Education (years)	10.34	1.85	9.53	1.90	*t* (68) = 1.82	0.073
Duration of methamphetamine abstinence (days)	-		264.89	167.35	-	-
Lifetime methamphetamine dosage (grams)	-		265.57	428.87	-	-
DSM-5 (methamphetamine)	-		8.43	2.05	-	-

**Table 2 brainsci-12-00974-t002:** Questionnaire results.

	HC (N = 35, Female = 17)		MUD (N = 35, Female = 17)		*t* Value	*p* Value
	Mean	SD	Mean	SD	(*df* = 68)	
BDI-II (depression)	9.86	9.45	11.82	9.46	−0.86	0.391
STAI-T (trait anxiety)	57.37	5.41	56.26	8.86	0.64	0.528
Sensation Seeking Scale	13.54	5.87	16.86	5.10	−2.52	0.014
Boredom Susceptibility	1.97	1.56	2.20	1.43	−0.64	0.525
Disinhibition of Desire	2.83	1.93	4.00	2.61	−2.13	0.037
Experience Seeking	3.89	1.55	4.11	1.68	−0.59	0.555
Thrill and Adventure Seeking	4.86	2.65	6.54	2.16	−2.92	0.005

**Table 3 brainsci-12-00974-t003:** Behavioral variables of MUD and HC participants.

	HC (N = 35, Female = 17)		MUD (N = 35, Female = 17)		*t* Value	*p* Value
	Mean	SD	Mean	SD	(*df* = 68)	
Certain-Go RT (ms)	428.37	56.17	415.71	41.22	-	-
Ignore-Go RT (ms)	435.54	50.87	425.97	41.28	-	-
Uncertain-Go RT (ms)	521.40	67.16	545.80	79.10	-	-
Fail-to-stop Go RT (ms)	459.43	50.42	468.34	63.80	-	-
Baseline RT (ms)	585.40	102.64	621.26	115.45	−1.37	0.174
Preparation cost (ms)	92.97	50.54	130.14	52.68	−3.01	0.004
Critical SSD (ms)	261.14	53.63	280.37	51.04	−1.54	0.129
SSRT (ms)	260.20	43.18	265.37	56.11	−0.43	0.667
PSS effect (ms)	44.31	33.25	36.43	37.51	0.93	0.355
Certain-Go accuracy	97.82%	3.00%	97.94%	2.60%	-	-
Ignore-Go accuracy	95.71%	4.10%	94.94%	4.44%	-	-
Uncertain-Go accuracy	97.50%	2.63%	97.14%	3.43%	-	-
Stop accuracy	49.94%	4.89%	51.25%	5.61%	-	-

## Data Availability

The data that support the findings of this study are available from the corresponding authors, Q.L. and X.L., upon reasonable request.

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
