# Peer review of "Patients with Methamphetamine Use Disorder Show Highly Utilized Proactive Inhibitory Control and Intact Reactive Inhibitory Control with Long-Term Abstinence"

_brainsci, 2022, doi:10.3390/brainsci12080974_

Round 1

Reviewer 1 Report

The article Patients with methamphetamine addiction show highly utilized proactive inhibitory control and intact reactive inhibitory control with long-term abstinence brings a new insight on the behavioral characteristics of proactive and reactive inhibitory control in MA patients. The originality and added value is visible. Structure of the paper is well organized, it is easy to read and comprehend. 

Results are well presented and discussed. Limitations of the study are well described and considered.

There was no issue detected when reading the work therefore it can be recommended for publishing. 

Author Response

Point: There was no issue detected when reading the work therefore it can be recommended for publishing.

Response: Thank you for your careful review of our manuscript during your busy schedule, and we appreciate your recognition of our current research.

Reviewer 2 Report

The manuscript by Dai etal offers interesting information on the behavioral impairments in subjects that have abstained from methamphetamine (MA) use.

The study is clearly outlined with information on the subjects and the details of behavioral measures. Statistical analysis is also clearly indicated.

I have a few minor critiques-

The authors included both male and female subjects. Did they look for any gender differences in the behavioral phenotypes? This becomes important as gender differences do exist in relapse to drug seeking.

In the introduction, in para 1, the authors could elaborate on the current therapies to reduce relapse or craving in subjects with MA disorders, and why conducting the current study would help with behavioral therapies.

Similarly, while limitations and conclusions are well written, it will be important to indicate the importance of the current findings to future behavioral therapies to treat subjects with MA use disorder.

Minor-

First sentence of the introduction, please insert (MA) after methamphetamine and change chronic relapse disorder to chronic relapsing disorder.

There are places in the manuscript where methamphetamine is still spelled out and they can be replaced with MA. E.g. line 118.

Author Response

Thank you for your review report. The comments were very helpful in revising and improving our manuscript. We carefully studied the comments and hope that our responses fully meet the reviewers’ expectations. We also responded to each comment point-by-point below. All the changes are highlighted in RED in the manuscript.

Point 1: The study is clearly outlined with information on the subjects and the details of behavioral measures. Statistical analysis is also clearly indicated. The authors included both male and female subjects. Did they look for any gender differences in the behavioral phenotypes? This becomes important as gender differences do exist in relapse to drug seeking.

Response 1: Thank you for the careful review and positive comments regarding our work. Gender differences are observed in drug response, addictive behavior and mental health of MA patients (Corsi et al., 2014; Dluzen & Liu, 2008; Simpson et al., 2016). The study of gender differences is indeed an important perspective for understanding addiction mechanisms. To address this issue, we have added content on gender differences in inhibitory control of MA patients in the Methods, Results and Limitations section.

In terms of proactive inhibitory control, there is a lack of gender differences in previous studies of methamphetamine use disorders. Based on your suggestion, we explored gender differences in proactive inhibitory control for the current data. The results suggest that there is no gender difference in the behavioral phenotypes of proactive inhibitory control in either MA patients or HCs. Both male and female MA patients utilized more proactive inhibitory control compared to HCs. In terms of reactive inhibitory control, the current study found weak gender differences, with females slightly better than males. Despite this, no main effect of group nor the interaction of group and gender was found for SSRT. This indicates that there is only gender difference in reactive inhibitory control, but there is no gender difference of MA patients in reactive inhibitory control. The focus and greater interest of the current study is to explore the differences between MA patients and HCs to make some targeted treatment recommendations. We did not find gender differences in the inhibitory control of MA. This is consistent with previous studies of substance addiction using SST, where behavioral outcomes did not find significant interactions of group and gender (Luo et al., 2013; Smith et al., 2016; van der Plas et al., 2009). However, the sample size of the current study is relatively small, so more replicated studies as well as studies with expanded samples are needed in the future to further examine the reliability of this conclusion.

We added gender difference analysis in the Method section of the revised manuscript, as follows (line 221-222):

Two non-parametric two-way ANOVA (Scheirer-Ray-Hare Test) were used to investigate the effects of group and gender on preparation cost and SSRT by R.

We added gender difference analysis in the Result section of the revised manuscript, as follows (line 258-265):

To investigate gender differences in the inhibitory control of MUD, two non-parametric two-way ANOVA of preparation cost and SSRT were applied in current study. For preparation cost, we found a significant main effect of group, H (1) = 7.25, p = .007; however, the main effect of gender, H (1) = .04, p = .838, and the interaction between group and gender was not significant, H (1) = 1.98, p = .159. For SSRT, we found a significant main effect of gender, H (1) = 3.87, p = .049, male was worse than female; however, the main effect of group, H (1) = .07, p = .796, and the interaction between group and gender was not significant, H (1) = .51, p = .473.

We added gender difference analysis in the Limitations section of the revised manuscript, as follows (line 433-442):

Third, the current study did not find gender differences in inhibitory control of MUD. Although, gender differences are observed in drug response, addictive behavior and mental health of MA patients [61-63], a lack of gender differences of proactive inhibitory control previous studies of MUD. In current study, we did not find gender differences in inhibitory control of MUD, neither in proactive nor in reactive inhibitory control. This is consistent with previous studies of substance addiction using SST, where behavioral outcomes did not find significant interactions of group and gender [22,64,65]. However, the sample size of the current study is relatively small, so more replicated studies as well as studies with expanded samples are needed in the future to further examine the reliability of this conclusion.

Point 2: In the introduction, in para 1, the authors could elaborate on the current therapies to reduce relapse or craving in subjects with MA disorders, and why conducting the current study would help with behavioral therapies. Similarly, while limitations and conclusions are well written, it will be important to indicate the importance of the current findings to future behavioral therapies to treat subjects with MA use disorder.

Response 2: Sorry for our unclear expression. It' s indeed true that our study cannot provide an effective treatment protocol. The current study was only able to reveal whether there are abnormalities in proactive and reactive inhibitory control in MA patients with long-term abstinence, which could provide empirical evidence for future targeted treatment protocols. So, in the revised version we changed “Therefore, investigating the inhibitory control of MA patients is vital to understand the mechanisms underlying addiction and to recommend effective treatments.” to “Therefore, investigating the inhibitory control of MA patients is vital to understand the mechanisms underlying addiction” (line 26-28).

In addition, we added the implications of the results of this study for future treatment of methamphetamine addiction and supplement them in the Limitation and prospect section. The details are as follows (line 443-453):

The difficult recovery of proactive inhibitory control may be an important reason for the high relapse rate of addiction. Previous studies have found that reactive inhibitory control could be recovered with long-term abstinence therapy [22-24], but the relapse rate after abstinence remains high [66,67]. The current study found no difference in reactive inhibitory control between MA with long-term abstinence and HCs, but proactive inhibitory control still remained a group difference. It suggests that long-term abstinence treatment may have restored reactive inhibitory control, but the improvement of proactive inhibitory control is still insufficient, which may be an important factor for the high the relapse rate. These results suggest that the future treatment of addiction should be further developed to treat proactive inhibitory control. Through training proactive inhibition, make it become the habitual response of MA patients, in order to reduce the relapse rate.

Point 3: First sentence of the introduction, please insert (MA) after methamphetamine and change chronic relapse disorder to chronic relapsing disorder. There are places in the manuscript where methamphetamine is still spelled out and they can be replaced with MA. E.g. line 118.

Response 3: Thank you for your suggestion. We have changed “chronic relapse disorder” to “chronic relapsing disorder” (line 17). As suggested by another reviewer, in order to strictly comply with the DSM-V definition, we changed the term “methamphetamine addiction (MA)” to “methamphetamine use disorder (MUD)” throughout the text. At line 112 (line 118 in first version), our expression may be not precise, and to reduce ambiguity, we change “abstained from methamphetamine use” to “abstained from methamphetamine intake”.

References

Corsi, K. F., Garver-Apgar, C., & Booth, R. E. (2014). Gender differences in HIV risk and mental health among methamphetamine users. Drug and Alcohol Dependence, 140, e39. doi:http://doi.org/10.1016/j.drugalcdep.2014.02.127

Dluzen, D. E., & Liu, B. (2008). Gender differences in methamphetamine use and responses: A review. Gender Medicine, 5(1), 24-35. doi:http://doi.org/10.1016/S1550-8579(08)80005-8

Luo, X., Zhang, S., Hu, S., Bednarski, S. R., Erdman, E., Farr, O. M., . . . Li, C.-s. R. (2013). Error processing and gender-shared and -specific neural predictors of relapse in cocaine dependence. Brain, 136, 1231-1244. doi:http://doi.org/10.1093/brain/awt040

Simpson, J. L., Grant, K. M., Daly, P. M., Kelley, S. G., Carlo, G., & Bevins, R. A. (2016). Psychological Burden and Gender Differences in Methamphetamine-Dependent Individuals in Treatment. Journal of psychoactive drugs, 48(4), 261.

Smith, J. L., Iredale, J. M., & Mattick, R. P. (2016). Sex differences in the relationship between heavy alcohol use, inhibition and performance monitoring: Disconnect between behavioural and brain functional measures. Psychiatry Research-Neuroimaging, 254, 103-111. doi:http://doi.org/10.1016/j.pscychresns.2016.06.012

van der Plas, E. A., Crone, E. A., van den Wildenberg, W. P., Tranel, D., & Bechara, A. (2009). Executive control deficits in substance-dependent individuals: a comparison of alcohol, cocaine, and methamphetamine and of men and women. J Clin Exp Neuropsychol, 31(6), 706-719. doi:http://doi.org/10.1080/13803390802484797

Reviewer 3 Report

This study by Weine Dai and colleagues investigated reactive and proactive inhibitory control in people with methamphetamine use disorder with long-term abstinence using a modified stop signal task, and their relationship with sensation seeking. They found that proactive inhibitory control was low compared to healthy controls, whereas there were no differences in reactive inhibitory control. There was also a correlation between sensation seeking and inhibitory control.

Overall, the manuscript is well written and offers novel insight into the neurocognitive processes underlying methamphetamine use disorder, as the authors tease apart different aspects of inhibition. Strengths include a solid introduction and well-established aims and hypotheses. They also use individual data point graphs and violin plots, which do a great job at highlighting individual variability in their sample. The manuscript could however benefit from some methodological clarification and use of more appropriate language. Please find below specific comments:

Introduction:

1.     Title, Line 17, Line 78 etc.: If the authors are going to use the DSM-5 definition, “methamphetamine addiction” must be replaced with “stimulant use disorder – methamphetamine” or “methamphetamine use disorder”. Pease update throughout the manuscript.

2.      Line 25: please use the politically correct term “people who use methamphetamine” or “people with methamphetamine use disorder” rather than “methamphetamine abusers”. Please correct throughout the manuscript

3.     Line 20-22: I would suggest citing recent reviews showing that inhibitory control is impaired in people with methamphetamine use disorder (e.g. PMID 31920743; PMID 29407687).

4.      Lines 47-51: literature relating to brain structure and function should be moved to the discussion, as the present paper does not investigate it and the introduction is already lengthy

5.      Lines 53-57: please provide a citation for this statement

6.      Line 97: I believe the authors meant sensation seeking instead of sensory? Please correct

7.     Line 101: please use politically correct term “people with cannabis use disorder”

Methods:

8.     Lines 113-115: if the authors are using the DSM-5 criteria, then the inclusion criteria should be “no substance use disorder” rather than “no addictions”.

9.    Line 113-115: please clarify if this includes cannabis use disorder. Cannabis and methamphetamine use are often co-occurring, and I wonder if the authors have excluded participants on the basis of cannabis use. If not, generalizability of data should be discussed in the limitations section

10.  Line 116: please clarify if they met mild, medium or severe criteria

11.  Line 119: “had previously taken an average dose of 265.57 (SD = 428.87) grams”. Is this lifetime dose? Please clarify

12.   Line 120: please clarify was is meant by “centres”. More information about where controls were recruited is needed.

13.  Line 123: “if they met any criteria for abuse or dependence” – substance abuse and dependence are DSM-IV criteria. Were healthy controls assessed based on the DSM-IV and the patients assessed with DSM-5? Please clarify and/or correct accordingly.

14.  Line 124: Was the incentive paid in cash or in the form of gift cards? Please clarify

15.   Table 1: In the text (Lines 113-115) it is stated the participants were excluded based on addiction to other substances including heroin, however the table lists mean DSM-5 heroin use. Please clarify whether participants were or were not excluded if they used heroin. In addition, it is unclear what the “mean = 8.43” figure refers to? It would be better to just state the number of participants meeting DSM-5 criteria for heroin use disorder, and perhaps the proportion. Lastly, please consider whether heroin use should be included as covariate in the analyses.

16.   Table 1: it is unclear what historic dose of methamphetamine means. Did the authors mean lifetime?

Results:

17.   It would be interesting to see the correlation between lifetime methamphetamine use and inhibitory control measures. 

Discussion:

19.   Lines 289-292: please cite this statement

20.   Lines 302-304: please cite this statement

21.   Line 314: “may be related to aberrant brain function caused by long-term methamphetamine use”. Can the authors go into more details as to which particular brain regions may be affected by long term methamphetamine use, and how it relates to inhibitory control?

22.   Line 320-322: It should be clarified in the method section (2.1. Participants) that all participants underwent treatment targeting inhibition and avoidance.

23.   Line 333-335: consider citing these papers showing a link between inhibition and cue-induced craving in people who use methamphetamine: PMID 22257306; PMID 21451018; PMID 34385074.

References:

Guerin AA, Drummond KD, Bonomo Y, Lawrence AJ, Rossell SL, Kim JH. Assessing methamphetamine-related cue reactivity in people with methamphetamine use disorder relative to controls. Addict Behav. 2021 Dec;123:107075. doi: 10.1016/j.addbeh.2021.107075. Epub 2021 Jul 31. PMID: 34385074.

Guerin AA, Bonomo Y, Lawrence AJ, Baune BT, Nestler EJ, Rossell SL, Kim JH. Cognition and Related Neural Findings on Methamphetamine Use Disorder: Insights and Treatment Implications From Schizophrenia Research. Front Psychiatry. 2019 Dec 17;10:880. doi: 10.3389/fpsyt.2019.00880. PMID: 31920743; PMCID: PMC6928591.

Potvin S, Pelletier J, Grot S, Hébert C, Barr AM, Lecomte T. Cognitive deficits in individuals with methamphetamine use disorder: A meta-analysis. Addict Behav. 2018 May;80:154-160. doi: 10.1016/j.addbeh.2018.01.021. Epub 2018 Jan 31. PMID: 29407687.

Tabibnia G, Monterosso JR, Baicy K, Aron AR, Poldrack RA, Chakrapani S, Lee B, London ED. Different forms of self-control share a neurocognitive substrate. J Neurosci. 2011 Mar 30;31(13):4805-10. doi: 10.1523/JNEUROSCI.2859-10.2011. PMID: 21451018; PMCID: PMC3096483.

Author Response

Thank you for your review report. The comments were very helpful in revising and improving our manuscript. We carefully studied the comments and hope that our responses fully meet the reviewers’ expectations. We also responded to each comment point-by-point below. All the changes are highlighted in RED in the manuscript.

Point 1: Title, Line 17, Line 78 etc.: If the authors are going to use the DSM-5 definition, “methamphetamine addiction” must be replaced with “stimulant use disorder – methamphetamine” or “methamphetamine use disorder”. Pease update throughout the manuscript.

Response 1: As your suggestion, in order to strictly comply with the DSM-V definition, we have changed the term “methamphetamine addiction (MA)” to “methamphetamine use disorder (MUD)” throughout the text.

Point 2: Line 25: please use the politically correct term “people who use methamphetamine” or “people with methamphetamine use disorder” rather than “methamphetamine abusers”. Please correct throughout the manuscript

Response 2: Sorry for incorrect wording. We have changed the “methamphetamine abusers” into “people who use methamphetamine” (line 25).

Point 3: Line 20-22: I would suggest citing recent reviews showing that inhibitory control is impaired in people with methamphetamine use disorder (e.g. PMID 31920743; PMID 29407687).

Response 3: Thank you for the two recent reviews of MUD inhibitory control. We have cited them in the corresponding places (line 72), providing stronger evidences.

Point 4: Lines 47-51: literature relating to brain structure and function should be moved to the discussion, as the present paper does not investigate it and the introduction is already lengthy.

Response 4: Following your suggestion, we have moved this paragraph on the similarities and differences in brain mechanisms of proactive and reactive inhibitory control to the Discussion section (line 404-409, 413-416). It provides a more theoretical basis for the theory of dual mechanisms of cognitive control.

Point 5: Lines 53-57: please provide a citation for this statement.

Response 5: Sorry for the missing citation, we have cited it in line 51.

Point 6: Line 97: I believe the authors meant sensation seeking instead of sensory? Please correct.

Response 6: Sorry for misuse. We have corrected it in line 90.

Point 7: Line 101: please use politically correct term “people with cannabis use disorder”

Response 7: Sorry for incorrect wording. We have changed the “cannabis- and gambling-addicted patients” into “people with cannabis use disorder or pathological gambling problem” (line 94).

Methods:

Point 8: Lines 113-115: if the authors are using the DSM-5 criteria, then the inclusion criteria should be “no substance use disorder” rather than “no addictions”.

Response 8: Sorry for the inappropriate description. We have corrected it in line 106-107.

Point 9: Line 113-115: please clarify if this includes cannabis use disorder. Cannabis and methamphetamine use are often co-occurring, and I wonder if the authors have excluded participants on the basis of cannabis use. If not, generalizability of data should be discussed in the limitations section.

Response 9: Sorry for the missing report on cannabis use disorder. All the MUD patients are primarily monosubstance-dependent users. We have added this information in the corresponding places (line 107)

Point 10: Line 116: please clarify if they met mild, medium or severe criteria

Response 10: We add the description in line 111-112: “All the MUD patients are primarily monosubstance-dependent users and met the severe criteria.”

Point 11: Line 119: “had previously taken an average dose of 265.57 (SD = 428.87) grams”. Is this lifetime dose? Please clarify.

Response 11: Yes, this average dose is the lifetime dose. We add a more detailed description in line 118-120, as following: “And each subject's lifetime methamphetamine dosage was estimated by recalling the sum of the daily dosage multiplied by the number of days used in each month from the first to the most recent methamphetamine use.”

Point 12: Line 120: please clarify was is meant by “centers”. More information about where controls were recruited is needed.

Response 12: Sorry for using an ambiguous description. We changed this sentence to “Thirty-five HCs from nearby communities were recruited through referrals from center staff, word-of-mouth, and print advertisements.” (line 121-122)

Point 13: Line 123: “if they met any criteria for abuse or dependence” – substance abuse and dependence are DSM-IV criteria. Were healthy controls assessed based on the DSM-IV and the patients assessed with DSM-V? Please clarify and/or correct accordingly.

Response 13: Sorry for the inappropriate description. Both of MUD patients and HCs used the DSM-V. We changed “if they met any criteria for abuse or dependence” to “if they met any criteria for substance use disorders” (line 124-125).

Point 14: Line 124: Was the incentive paid in cash or in the form of gift cards? Please clarify.

Response 14: We clarified in line 126-127, as following: “The incentive was transferred to MA patients' savings cards and to HCs' Alipay accounts.”

Point 15: Table 1: In the text (Lines 113-115) it is stated the participants were excluded based on addiction to other substances including heroin, however the table lists mean DSM-5 heroin use. Please clarify whether participants were or were not excluded if they used heroin. In addition, it is unclear what the “mean = 8.43” figure refers to? It would be better to just state the number of participants meeting DSM-5 criteria for heroin use disorder, and perhaps the proportion. Lastly, please consider whether heroin use should be included as covariate in the analyses.

Response 15: Sorry for making such a big mistake. The item for DSM-V in Table 1 should be “DSM-V (methamphetamine)”. All the MUD patients are primarily monosubstance-dependent users. “mean = 8.43” means the average of DSM-V (methamphetamine) scores for all MUD patients.

Point 16: Table 1: it is unclear what historic dose of methamphetamine means. Did the authors mean lifetime?

Response 16: Yes, it means lifetime methamphetamine dosage. We have corrected it in table 1.

Results:

Point 17: It would be interesting to see the correlation between lifetime methamphetamine use and inhibitory control measures.

Response 17: This is really good advice. We tried to analyze the correlation between lifetime methamphetamine use and inhibitory control measures. No significant correlation was found between lifetime methamphetamine use and preparation cost (r = -.04, p = 1.000) and between lifetime methamphetamine use and SSRT (r = .06, p = 1.000). The reason for the absence of results may be because the measurement of lifetime methamphetamine use was self-reported (uncertainty of reality), retrospective (memory bias), and therefore may not be accurate enough.

Discussion:

(Perhaps due to typographical problems, there is no number 18 point in review report 3. In order to prevent the wrong correspondence of questions, we use the original point number in the response letter.)

Point 19: Lines 289-292: please cite this statement

Response 19: Sorry for the missing citation, we have cited it in line 306.

Point 20: Lines 302-304: please cite this statement

Response 20: Sorry for the missing citation, we have cited it in line 318.

Point 21: Line 314: “may be related to aberrant brain function caused by long-term methamphetamine use”. Can the authors go into more details as to which particular brain regions may be affected by long term methamphetamine use, and how it relates to inhibitory control?

Response 21: We add the following description to the Discussion section (line 329-336): “Previous studies of inhibitory control have found that right inferior frontal gyrus (rIFG), anterior cingulate gyrus and prefrontal cortex activation were decreased in MUD patients who had been abstinent for more than 3 weeks during the Stroop task [18,38]. Moreover, the attenuated rIFG activation during proactive inhibitory control found in cocaine use disorders may imply that the blunted inhibitory ability of CUD participants stems from deficits in initial attention and detection of potential stopping targets [39]. However, there is a lack of brain imaging studies of proactive inhibitory control of MUD, as more fMRI studies on this issue are needed in the future.”

Point 22: Line 320-322: It should be clarified in the method section (2.1. Participants) that all participants underwent treatment targeting inhibition and avoidance.

Response 22: We add the following description to the Method section (line 114-118): “In abstinence treatment, the drug rehabilitation centers help MUD patients to suppress/stay away from addiction-related stimuli by conducting expert lectures, popularization of science and speaking on behalf of successful drug rehabilitation, which benefit MUD patients' awareness of the dangers of drugs and their initiative to stay away from drugs.”

Point 23: Line 333-335: consider citing these papers showing a link between inhibition and cue-induced craving in people who use methamphetamine: PMID 22257306; PMID 21451018; PMID 34385074.

Response 23: Thank you for your suggestion. We have cited them in the corresponding places (line 357), providing stronger evidences.
